# Tooth position prediction method based on adaptive geometry optimization

**Tian Ma**<sup>◔</sup>, **Yijie Zeng** ◔*, **Wenda Pei**‡, **Chao Li**, **Yuancheng Li**

College of Artificial Intelligence & Computer Science, Xi'an University of Science and Technology, Xi'an Shaanxi, China

‡ These authors also contributed equally to this work.
◔ These authors contributed equally to this work.

* 1047843612@qq.com

**Data availability statement:** All data files are available from the URL: https://gitcode.com/zyj_jj/TPPMAGO_toothdata/blob/main/README.md database. All code files are available from URL: https://gitcode.com/zyj_jj/TPPMAGO.

## Abstract

A multi-layer feature optimization Transformer-based tooth position prediction method is proposed to address the problems of difficult access to high-precision medical data and the difficulty of capturing and representing hierarchical features and spatial relationships among teeth by current methods. First, a geometric adaptive optimization strategy and a physiological adaptive reconstruction strategy are designed for real-time adaptation to the complexity of different clinical environments and enhanced pose invariance by integrating the physiological characteristics and anatomical structure of teeth. Then, a hierarchical feature tooth position prediction network was designed to solve the problems of weak ability of MLPs to process high-dimensional data and low accuracy of prediction transformation matrix by extracting hierarchical geometric features of teeth. Finally, a jointly supervised loss function is constructed, which can simultaneously capture the intrinsic differences, spatial relationships and uncertainties of the tooth position prediction disorder distribution, and can effectively supervise the tooth spatial structure relationships and prevent tooth collisions and misalignments. The experimental results show that the accuracy of the proposed method is improved by 2.87% and the rotation and translation errors are reduced by 28.28% and 37.53%, respectively, compared with the current method.

## Introduction

Dental position prediction has attracted great attention in recent years [1], Malocclusion is a common and multiple oral disease [2], whose main symptoms are misalignment of teeth and facial deformity [3]. Tooth position prediction is dedicated to the study and treatment of structural abnormalities of teeth [4]. MOTOHASHI and KURODA [5] performed tooth alignment by manual method in 1999, this process is labor intensive and time consuming. Therefore, learning based digital orthodontics has become an emerging field [6].

Determining the ideal tooth position is an important step in digital orthodontics [7]. where the malocclusion tooth position is used as the initial position and the ideal tooth position is used as the alignment target, which is usually performed by a physician and is highly

**Funding:** This work was supported by the Key Technologies Research and Development Program (No. 2022ZD0119005), awarded to TM, andinpart by the Natural Science Foundation of Shaanxi Province (No. 2022JM-508), awarded to TM. TM played a key role in research design, data collection, manuscript preparation, and decisions regarding publication.

**Competing interests:** The authors have declared that no competing interests exist.

subjective [8]. Therefore, a tooth position prediction method is needed to reduce misclassification [9]. TANet [10] is the first learning-based method for predicting tooth alignment targets, which uses PointNet [11] for crown point cloud feature extraction, and Graph Neural Networks [12] for feature propagation between teeth and predicting ideal positions.TAligNet is a lightweight model based on PointNet proposed in [13] but MLPs lack the ability to deal with complex spatial relationships.PSTN target physiological datasets through IDR and ITD [14], but IDR and ITD do not adequately take into account the physiological properties of the teeth in the above strategies [15].

In summary, this paper presents an end-to-end tooth position prediction method by formulating the tooth position prediction task as a structured 6D attitude estimation task. The main contributions are as follows:

(1) Geometric Adaptive Optimization Strategy (GAOS) and Physiological Adaptive Reconstruction Strategy (PARS) are proposed. The problem of scarcity of high-quality medical data is overcome, and the adaptive ability on the physiological diversity of model teeth is optimized.

(2) The Feature-Hierarchical Vector Attention Multi-Layer Prediction Network (F-VAMP) network was designed to overcome the problem that MLPs are sensitive to the order in which the input tooth data are arranged and cannot fully understand the overall structure of the teeth and the tooth relative relationships between them.

(3) A new loss function is designed that can simultaneously provide effective supervision of tooth position prediction by capturing the intrinsic differences, spatial relationships, and uncertainties in the distribution of tooth position prediction disorders.

## Related work

The problem of pose estimation has been extensively studied in recent years, which aims to infer the 3D pose of an object present in an RGB image [16], RGB- d image [17] or point cloud data [18] that has six degrees of freedom. Existing methods can be broadly categorized into target coordinate regression methods and template matching methods. Coordinate regression-based methods estimate the corresponding object surface for each object at the pixel level, assuming that the corresponding 3D model is known for training [19]. Template matching based methods use various techniques such as iterative closest point (ICP) to align known 3D models with image observations [20]. In 2020, Li et al. [20] proposed assembling 3D parts guided by additional images but did not explicitly consider the connection points between parts. In 2020, Huang et al. [21] constructed a dynamically varying part map to induce six-degree-of-freedom poses, but the performance in six-degree-of-freedom pose prediction is still not very satisfactory. They usually focus on the design of the relational inference network architecture and ignore the instance differences among the parts. Taking a chair as an example, on the one hand, although some parts share the same geometry, they predict different bit postures in practice. Geometrically identical input parts usually lead to intra-class conflicts and fail to accurately predict different postures.

In recent years, research on dental position prediction based on deep learning has gradually appeared. In 2020, Li et al. [14] developed a deep learning tooth position prediction framework based on PSTN and proposed two new IDR and ITD strategies, which mainly focus on the changes in tooth morphology to obtain the physiological dataset needed for tooth position prediction, but these two strategies do not fully consider the physiological characteristics and anatomical structure of teeth. In 2020, Wei et al. [10] developed TANet, a deep learning-based framework for automated tooth position prediction, which explicitly modeled the automated tooth alignment task as a structured 6D position prediction problem

for the first time, and learned the mapping relationship from the initial layout of teeth to the target layout from the clinical data through supervised learning, with the network containing a feature encoding module, a feature propagation module, and a position regression module. However, they improved the feature module and used only MLPs for prediction in the pose regression module, which does not use local correlation when dealing with unseen and complex information, and it usually does not perform as well as neural networks designed for point clouds, and the neural network-predicted tooth-targeted poses do not take into account more fine-grained geometrical features of the teeth, which need to be further processed [22]. In 2022, Chen et al. [9] identified important positions on the teeth by detecting the markers on each tooth, and used this to construct a feature map structure for jaws and teeth, and this structured representation was able to efficiently capture the complex topological relationships between the teeth and adjacent anatomical structures. The study was guided by the normal occlusal relationships of clinical teeth and actively guided the network to learn the standard spatial layout and position of the teeth, which improved the accuracy of the network's prediction of rigid positional shifts of the teeth. However, what they improved was the feature extraction module, where the pose regression module used only MLPs to extract the fused three layers of information to get the six-degree-of-freedom output, which lacked capturing the spatial relationship of the teeth and was too dependent on the sample data for the input tooth model.

Although existing studies have provided different solutions to the tooth position prediction problem, with many attempts and improvements in feature extraction, pose regression, and structural modeling, there are still some pressing challenges that need to be addressed. First, there are significant individual differences among teeth, such as shape, size, and alignment, which may vary from person to person, making the model perform instably when generalized to different individuals. Second, due to the high similarity of multiple teeth in geometric morphology, it is easy to trigger intra-class conflict, i.e., the model is difficult to accurately distinguish the subtle differences between similar teeth, leading to biased predictions. In addition, some current methods are insufficient in dealing with the spatial arrangement of teeth and anatomical constraints, making it difficult to comprehensively capture the spatial interactions and occlusal characteristics of teeth, thus affecting the accuracy and stability of the overall prediction. Therefore, a new approach is proposed in this paper. Unlike PSTN [14], Chen [9] and TANet [10], this paper firstly proposes two novel strategies, GAOS and PARS, to enhance the feature extraction capability, model adaptation to complex feature inputs, and pose invariance. Secondly, the F-VAMP network is designed to improve the modeling and decoupling ability of the model to advanced feature information, and more accurately predict the dental ideal attitude. Finally, a jointly supervised loss function was designed to effectively supervise the spatial structural relationship of teeth and prevent tooth collision and misalignment.

## Physiological dynamic optimization strategy

### Geometric adaptive optimization strategy

By combining the global geometric properties with local features of point cloud data, this strategy aims to optimize tooth positioning with high precision during orthodontic treatment. This strategy innovatively introduces a multidimensional feature alignment and sampling consistency algorithm, which ensures the precise matching of the source and predicted point clouds in terms of geometrical structure while extracting the intermediate state of the dataset as the required physiological dataset, which significantly improves the processing efficiency of the physiological dataset.

The real-time computation of geometric centers, combined with the singular value decomposition technique, enables this method to extract high-dimensional rotation and displacement parameters, and use them to construct the local transformation matrix. On this basis, the interpolation strategy is incorporated to optimize the point cloud layer by layer, which enables the model to cope with complex dental morphology and spatial changes.

In order to break through the bottleneck problem in the traditional rotational representation, the present method adopts quaternions to construct the rotational transformation and extracts the rotational information from the transformation matrix, which effectively avoids the gimbal lock problem, and at the same time drastically reduces the complexity of storage and computation. By this method, the output of this strategy has excellent spatial transformation invariance while guaranteeing geometric consistency. The specific operation of GAOS is shown below:

As shown in Fig 1, three sets of result plots of the present strategy are shown, respectively. This strategy notates the source point cloud of the malocclusion as $X_{start}$, Record the pairwise corrected source point clouds as $X_{end}$, the global transformation matrix with respect to the world coordinate system is denoted as $A_{ori}$, the validation reveals that the significant difference in the number of point clouds between $X_{start}$ and $X_{end}$ adversely affects the accurate solution of the local transformation matrix and the validity of the resulting augmented data. For this reason, this strategy samples the data of $X_{start}$ and $X_{end}$ to ensure the consistency of the number of point clouds and thus obtain a standardized corrected dataset.

Algorithm 1 presents the core pseudo-code of the strategy. first, the inputs are the paths corresponding to the data of $X_{start}$ and $X_{end}$. Lists $list_p$ and $list_u$ are initialized for storing the conforming and non-conforming paths, respectively. Second, the geometric centers $\bar{x}$ and $\bar{y}$ of $X_{start}$ and $X_{end}$ are computed and the covariance matrix $S = X_{start}^T \cdot X_{end}$ is constructed based on this. Decompose the covariance matrix by singular value solving $\mathcal{U}, \sum, V^T = svd\,(S)$ and output the $4 \times 4$ column-major order transformation matrix. After the local transformation matrix is constructed, an interpolation algorithm is used to gradually apply the matrix to each part of the point cloud data, and each part is transformed in the same way. The transformation matrix is finally characterized by quaternion form, which effectively avoids the

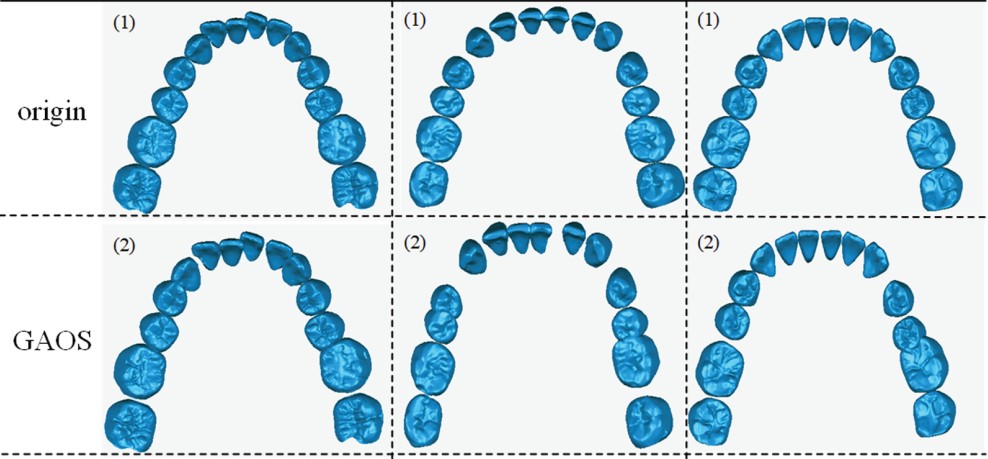

**Fig 1. Effect of GAOS strategy to generate dataset.** The top row represents the original dataset, while the bottom row illustrates the visualization effect of GAOS.

**Algorithm 1** Pseudocode for GAOS

**Require:** $Path_{start}(X_{start})$ and $Path_{end}(X_{end})$
**Ensure:** $matrix_g$ and $data_g$
 1: initialize lists $list_p$ and $list_u$
 2: $index \Leftarrow 1$
 3: **while** $index \leq len(Path_{start})$ **do**
 4:     $extract(X_{start})$ and $extract(X_{end})$
 5:     $Cen(X_{start}, X_{end})$
 6:     $S = X_{start}^T X_{end}$
 7:     $\mathcal{U}, \sum, V^T = svd(S)$
 8:     $= \mathcal{V} \mathcal{M} \mathcal{U}^T$
 9:     $= y - \bar{\mathcal{R}} \bar{x}$
 10:    local transformation matrix T
 11:    $index \Leftarrow index + 1$
 12: **end while**
 13: **if** $X'_{end} == X_{end}$ **then**
 14:     **if** $len(X_{start}) == len(X_{end})$ **then**
 15:         Added to $list_p$
 16:     **else**
 17:         Added to $list_u$
 18:     **end if**
 19: **end if**
 20: $index \Leftarrow 1$
 21: **while** $index \leq len(list_u)$ **do**
 22:     $i \Leftarrow 1$
 23:     **while** $i \leq len(Path_{start})$ **do**
 24:         Interpolated quaternions, displacements
 25:         augmentation data $X_{mid}$
 26:         $i \Leftarrow i + 1$
 27:     **end while**
 28:     $index \Leftarrow index_+ 1$
 29: **end while**

common gimbal lock problem in the traditional rotational representation, and also significantly reduces the computational complexity and time overhead of storage and transmission. Ultimately, the adaptive dental data $X_{mid}$ is generated by Eq (1), which is defined as follows:

$$X_{mid} = X_{start}^T \cdot T_r + T_t \tag{1}$$

where $T_r$ is the rotation transformation matrix and $T_t$ is the translation transformation matrix.

## Physiological adaptive reconstruction strategy

The PARS strategy integrates rotational and translational physiological constraints with complex geometric transformations to ensure that tooth movement in 3D space aligns with oral biomechanics. It generates clinically consistent physiological datasets based on patient data to better reflect real orthodontic conditions. By using a globally optimized Gaussian sampling method and 3D spatial affine transformations, PARS dynamically adjusts tooth positions and orientations, enabling the model to maintain geometric consistency and capture subtle spatial changes under varied clinical scenarios. The specific operation of PARS is shown below:

As shown in Fig 2, the outputs of the three sets of the present strategy are presented in the figure. This strategy proposes the constraint that the absolute tooth displacement is not greater than 10 mm, The randomly generated displacement vector $d_t$ generates a 3D translation vector $t = (t_x, t_y, t_z)$ in the coordinate space obeying a random distribution of $[-10,10]$, And the constraint that the absolute value is not greater than $10°$ is obeyed for selected x-, y- and z-axes in 3D space to reduce the probability of extreme cases. The computation of the local rotation matrix is calculated through Rodriguez's formula as shown in Eq (2):

$$r = I \cos\theta + (1 - \cos\theta)\, nn^T + \sin\theta\,[n] \tag{2}$$

where $\theta$ is the angle of rotation, I is the unit matrix, $[n]$ is the skew-symmetric matrix of the rotation axis vector n. The augmentation strategy is calculated as shown in Eq (3):

$$X_{mid} = rX_{start} + t \tag{3}$$

where the first 3∗3 positions of the transformation matrix T of length 4*4 form the rotation matrix r and the first 3 rows of the 4th column form the displacement matrix.

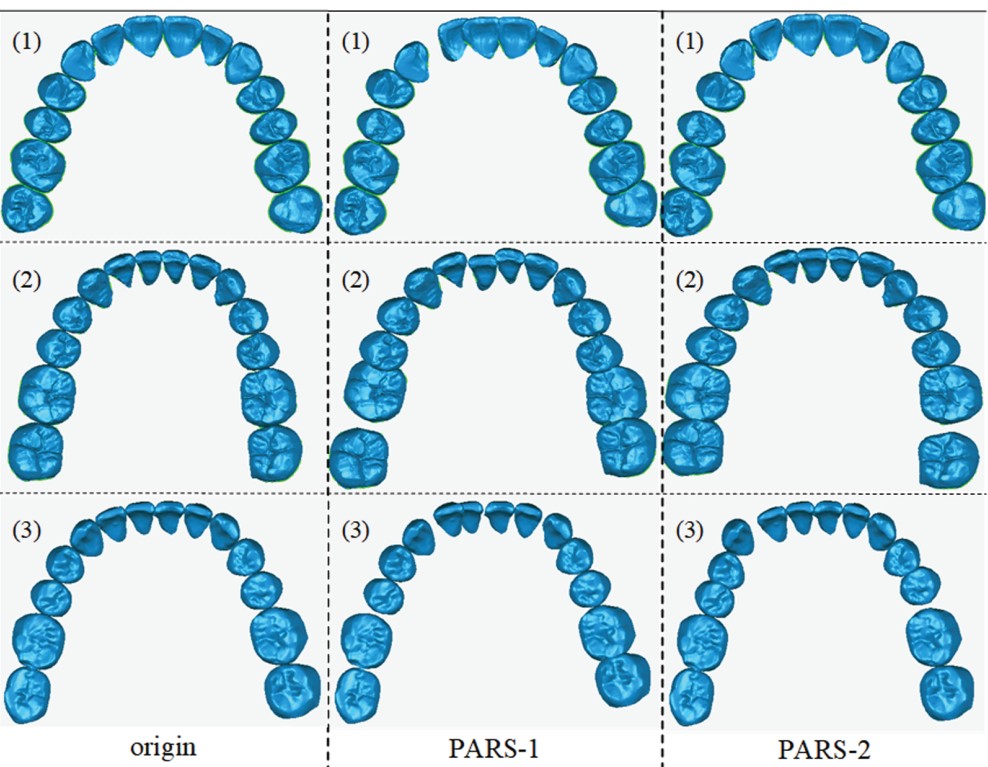

**Fig 2. Effect of PARS strategy to generate dataset.** The first column represents the original dataset, while the second and third columns illustrate the visualization effects of the PARS strategy.

## Presentation of the methodology

### Overall framework design

The overall framework design of the proposed method is shown in Fig 3, where the first module on the left is the Data Input Module(DT), which takes as input the triangular mesh model of the teeth before and after orthodontic treatment and extracts the point cloud data from it. The second module is the Hierarchical Feature Extraction Module (HFEM), which consists of five independent encoders: (1) the encoder Sampling Encoder is used to encode the tooth shape; (2) the encoder landmarks Encoder is used to encode the tooth landmarks; (3) the encoder Rim Encoder is used to encode the tooth axes; (4) the encoder Block Encoder for encoding tooth frames; (5) Encoder Convert Encoder for converting tooth axes to quaternions. The third module is the Ideal Tooth Posture Prediction Module(ITPPM), which takes the extracted tooth features as inputs and predicts the position of the ideal tooth through a hierarchical feature prediction network. The details of each module are described below.

### Hierarchical feature extraction module

Considering that the tooth position prediction task is usually limited by different features displayed on multiple scales, and that the position and orientation of the teeth may be affected by a variety of factors, including individual anatomical differences, misalignment or crowding of the teeth, the hierarchical features are shown in the blue box in Fig 3. The Sampling Encoder is used to encode the tooth shapes focusing on relatively localized features under the synergistic action of multiple teeth in order to capture the geometric features of the tooth model, and this method effectively avoids the computational complexity problem caused by directly processing the tooth point cloud or mesh data.landmarks Encoder focuses on capturing the descaling features as well as symmetry of each tooth, and the proposed method adopts the annotated tooth landmarks data to train and predict the landmarks corresponding to different types of teeth, and extracts the landmarks corresponding to each type of teeth shown in Fig 4, in which the cuts are shown in the blue box. The landmarks corresponding to each tooth type are extracted as shown in Fig 4, where the incisor landmarks contain Mesial and Distal, the cuspid landmarks contain Mesial and Distal, the premolar landmarks contain Mesial, Distal, Lip and Tongue and the posterior molar landmarks contain Mesial, Distal, MesialLip, DistantLip, MesialTongue and DistalTongue.The Rim Encoder and Block Encoder focus on the local and global features of the teeth. The fourth of these encoders interprets the distribution of each tooth in space through the OBB envelope box, while the Rim Encoder satisfies the requirement of orthogonality of the three axes of the tooth frame, treats these three axes as a whole, and represents this whole through quaternions, and this method is able to efficiently express the rotational attitude features of the tooth frame while maintaining the orthogonality. The proposed method migrates and fuses different tooth features as inputs to the tooth position prediction network to predict 6D posture for malocclusion.

### F-VAMP module

The F-VAMP tooth position prediction module significantly enhances the extraction and representation of high-dimensional features by incorporating advanced feature learning mechanisms. Specifically, this module enables each point in the point cloud to carry positional and other high-level features, allowing F-VAMP to process point cloud data effectively through positional encoding and feature embedding tailored for point clouds. Additionally, the property of focusing on the similarity between pixel points through the self-attention mechanism

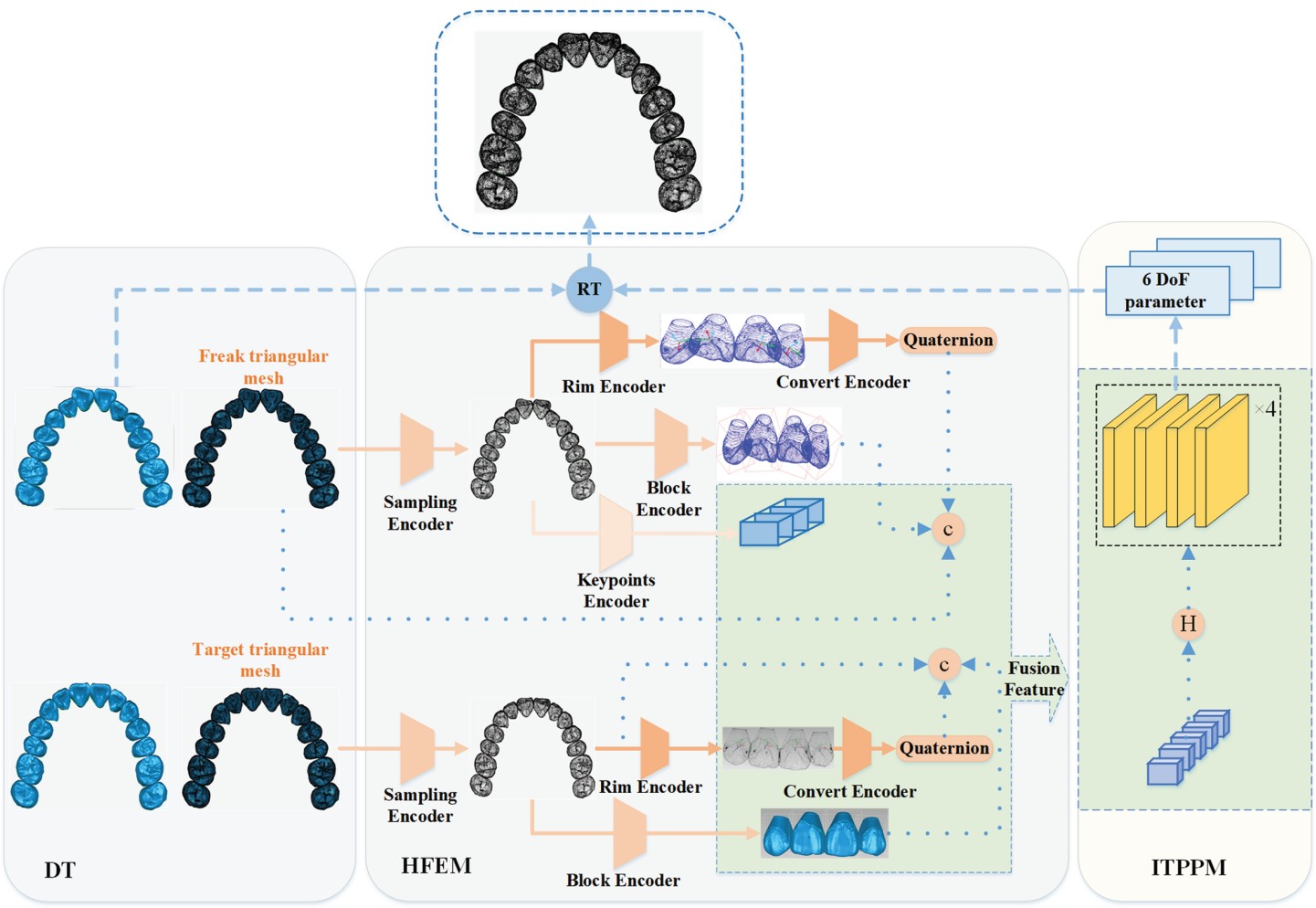

**Fig 3. Overall block diagram of tooth position prediction.** The module on the left is the Data Input Module, the module in the middle is the Hierarchical Feature Extraction Module, and the module on the right is the Ideal Tooth Posture Prediction Module.

maps to the need to dynamically assign weights to each transform matrix in this study, allowing the model to automatically focus on features related to the transform matrix, enhancing the ability to learn the similarity of the transform matrix. Finally, F-VAMP progressively enhances feature recognition by deeply capturing the high-level relationships among teeth extracted from the hierarchical network structure, combined with the application of One-Hot encoding.

In summary, the structure of F-VAMP is shown in Fig 5 as follows: (1) Nonlinear mapping and feature transformation of the input tooth potential representation features through the linear layer, as shown in Eq (4), where $Z_i$ is the nonlinear transformation, $x_i$ is the input feature, and $w_1$ and $b_1$ are the weights and biases of the linear transformation:

$$Z_i = W_1 x_i + b_1 \tag{4}$$

(2) The input features were encoded using feature coding to process the position and orientation information, respectively, and the position encoder and orientation encoder encoded

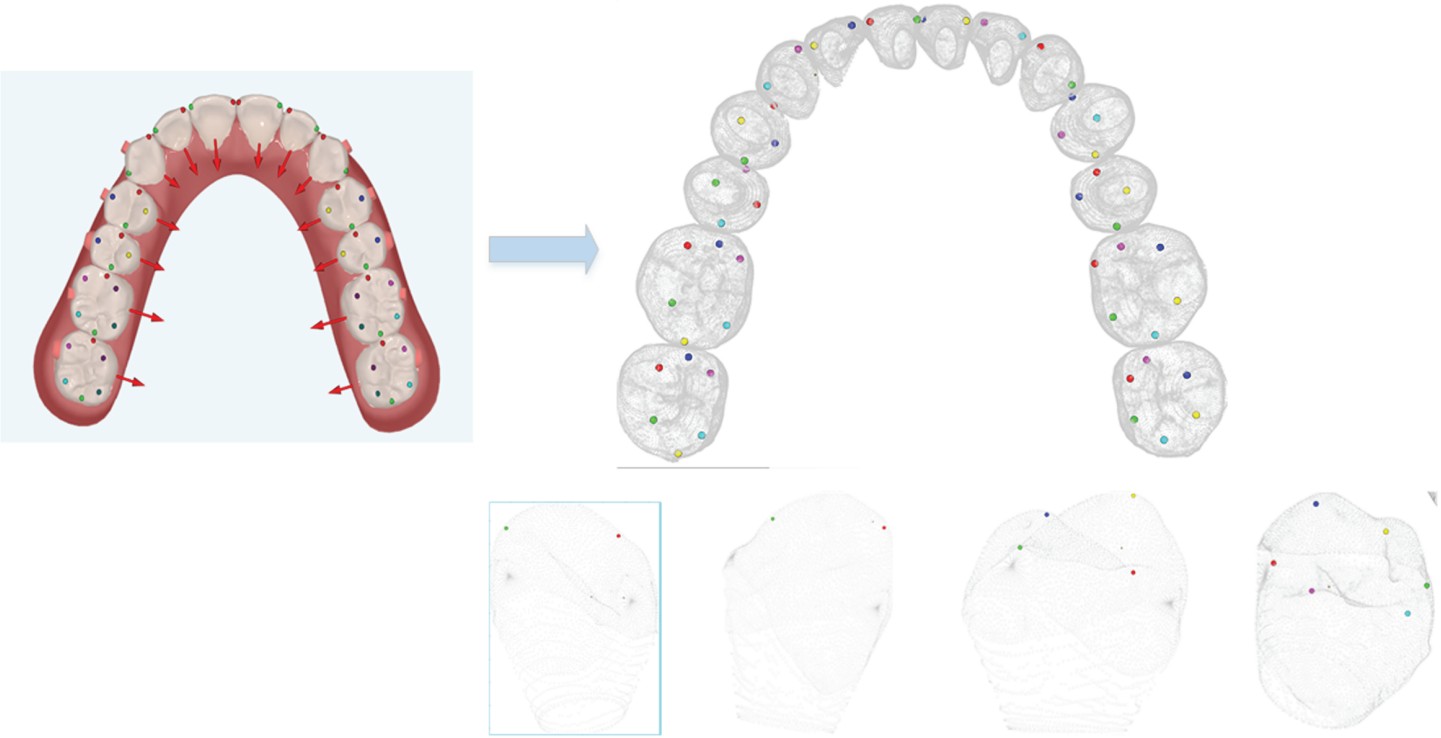

**Fig 4. Sketch of the key points of the tooth.** The points in different colors represent the key points for each type of tooth.

the input tooth landmarks $p_i$ and the tooth local coordinate system $d_i$ as features to generate higher dimensional feature vectors as shown in Eqs (5) and (6):

$$e_{pos} = PosEnc\,(p_i) \tag{5}$$

$$e_{dir} = PosEnc\,(d_i) \tag{6}$$

(3) The long distance dependence of the feature vectors is captured by the self-attention mechanism, which is important for the interrelationships between different teeth in tooth position prediction, as shown in Eq (7), where Z is the encoded feature vector and $Z'$ is the feature representation after processing by the self-attention mechanism:

$$Z' = Attention\,(Z) \tag{7}$$

(4) The processed high-level features of the teeth are subjected to feature migration and feature aggregation as shown in Eq (8), where $Z''$ is the feature after performing migration and aggregation.

$$Z'' = Aggregate\,(Z') \tag{8}$$

(5) The aggregated features are passed through a Multilayer Perceptron to output a 6D pose predicted for malocclusion as shown in Eq (9), where y is the 6D pose result.

$$y = MLP\,(Z'') \tag{9}$$

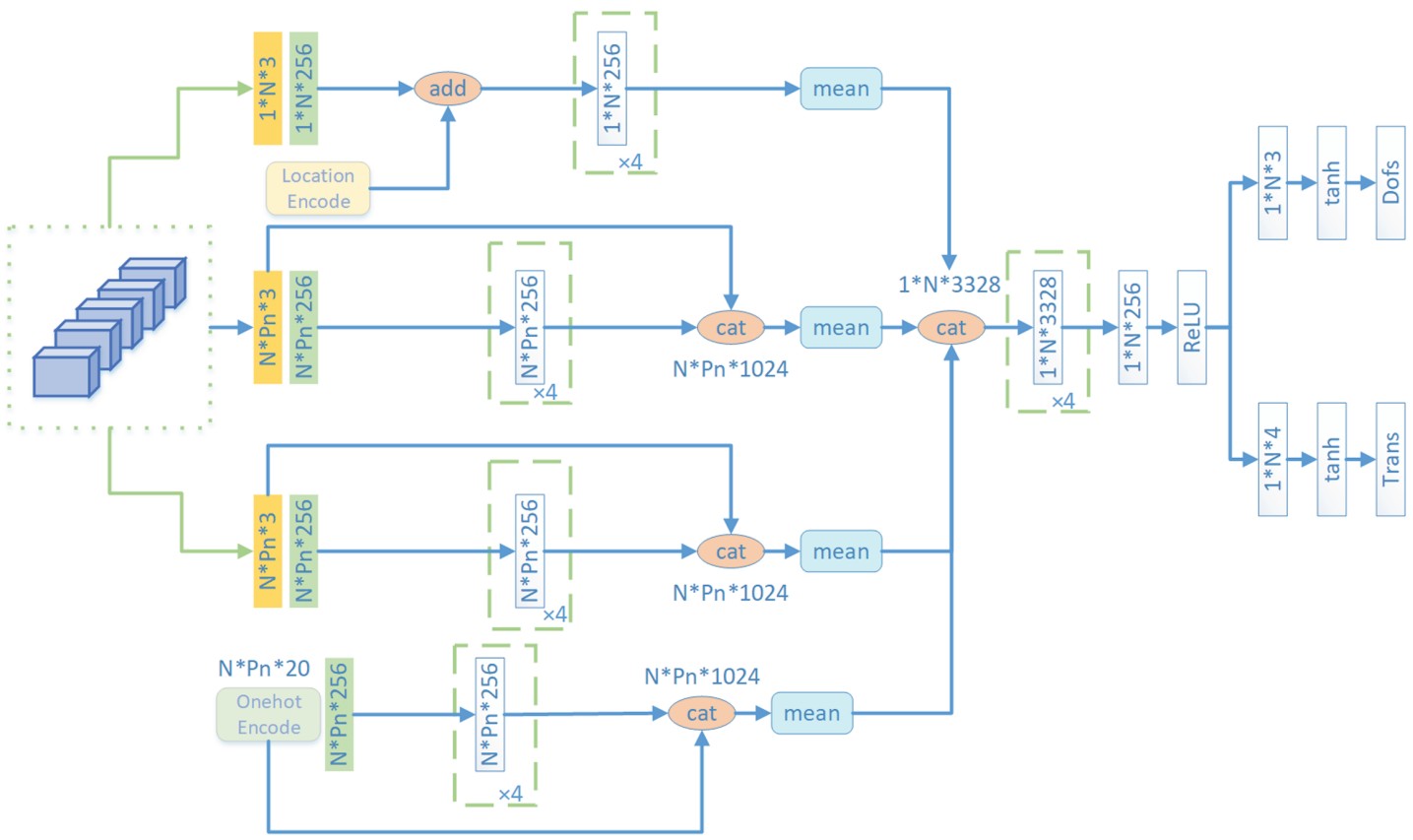

**Fig 5. F-VAMP multi-level tooth position prediction network model.** On the left, the input tooth data and the corresponding features are shown. After processing through the network layers, the output on the right represents the rotation and translation parameters for the ideal tooth position.

## Multi-objective loss function optimization module

In this method, a new joint loss function is designed, in which not only the accuracy of individual teeth is important in tooth position prediction, but also the relative positions between teeth are extremely critical. As shown in Fig 6a, the collision between teeth greatly affects the relative position location between teeth, and also seriously affects the accuracy of tooth position prediction.

Therefore, this method uses the grid point distance calculation method [9] and spatial core loss for preventing collisions and misalignments between teeth, as shown in Fig 6b. By minimizing the distance between predicted tooth grid points and real tooth grid surfaces, the model is able to learn the complex tooth morphology and details, and solve the problem of collision and misalignment between teeth. The grid distance calculation method mainly calculates the distance d between the source grid point $P_i$ and the target grid point $P_t$. The grid point distance loss $L_d$ is calculated as shown in Eq (10):

$$L_d = RMSE\left(\sum_{(P_i, P_t)\in k}\left[\frac{1}{\left(1+\frac{P_e}{P_c}\right)^{12}} - \frac{2}{\left(1+\frac{P_e}{P_c}\right)^{6}}\right]\right) \qquad (10)$$

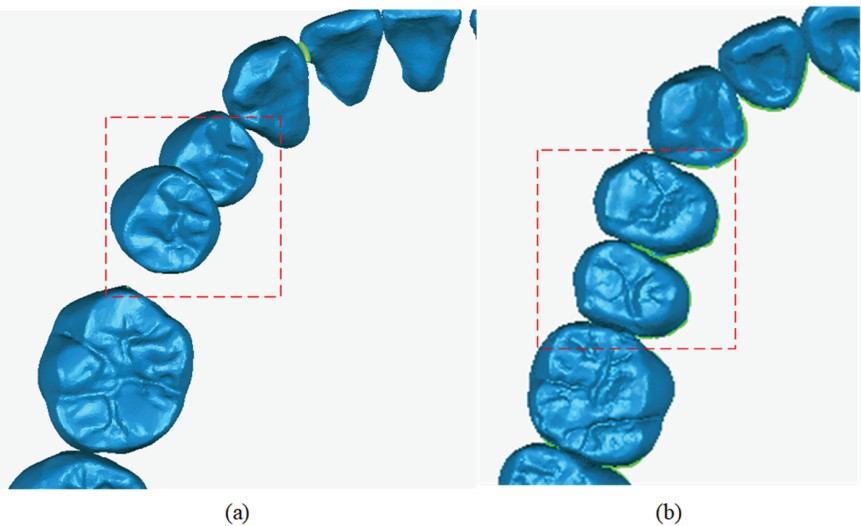

**Fig 6. Intertooth collision diagram.** Figure (a) shows the tooth collision diagram, while Figure (b) illustrates the effect of resolving the tooth collision issue.

where RMSE is the root-mean-square error, $C_e$ is the center-of-mass distance between teeth $P_i$ and $P_t$, and $P_e$ is calculated as shown in Eq (11):

$$P_e = P_e{'} + \sigma \tag{11}$$

where $P_e{'}$ represents the closest point-to-point distance between teeth $P_i$ and $P_t$, And by including the empirical parameter $\sigma$ to ensure that the minimum distances of $P_e$ do not overlap.

The spatial core loss $L_{core}$ is calculated as shown in Eq (12):

$$L_{core} = \frac{1}{n} \sum_{i=1}^{n} \| c_i - d_i \|^2 \tag{12}$$

where $c_i$ is the core location of the target and $d_i$ is the true core location.

The chamfer distance loss is a metric that measures the similarity between two point cloud collections and evaluates the difference between the two point clouds by calculating the average distance from each point in point set $P_1$ to the nearest point in point set $P_2$ and adding it to the average distance from each point in point set $P_2$ to the nearest point in point set $P_1$. The chamfer distance loss is a measure of the similarity between two point cloud collections. The chamfer distance loss $L_C$ is shown in Eq (13):

$$L_C = \frac{1}{n} \sum_{i=1}^{n} \left( \frac{1}{P_1} \sum_{p \in P_i} \min_{q \in Q_i} \| p - q \|^2 + \frac{1}{P_2} \sum_{q \in Q_i} \min_{p \in P_i} \| p - q \|^2 \right) \tag{13}$$

where n is the batch size.

Based on the observation that the teeth remain almost rigid during processing and the network of the proposed method also maintains the shape of each tooth in the input, the present approach maintains the prediction of the rigid body through the geometric reconstruction

loss $L_{build}$ to be consistent with the true value as shown in Eq (14):

$$L_{build} = \frac{1}{n} \sum_{i=1}^{n} \omega_i \left( \gamma(p - C(p, X_t[i]))^2 + \alpha(m(X_v[i]) - m(X_t[i]))^2 \right) \tag{14}$$

where $X_v$ denotes the predicted point cloud, $X_t$ denotes the target point cloud, $C(p, X_t[i])$ is the nearest point to p found in $X_t[i]$, $\gamma$ and $\alpha$ are the weighting parameters.

The gimbal lock problem is usually a very critical issue when it comes to the Euler angle rotation problem, so this method supervises the rotation transformation to avoid the gimbal lock problem as much as possible by means of the geometric metric loss $L_{dof}$, which combines the robustness of the $L_1$ loss to outliers and the efficient gradient property of the $L_2$ loss, which is computed as shown in Eq (15):

$$L_{dof} = \begin{cases} \frac{\sum_{i=1}^{n} \beta \left| p_{dofs,i} - g_{dofs,i} \right|^2 \cdot \omega_{rw,i}}{n}, & \left| p_{dofs,i} - g_{dofs,i} \right| < 1 \\ \sum_{i=1}^{n} \frac{\left| p_{dofs,i} - g_{dofs,i} \right| - \beta}{n}, & \left| p_{dofs,i} - g_{dofs,i} \right| \geq 1 \end{cases} \tag{15}$$

This method looks at the rotational transformation matrix as a whole, converts it to quaternions, and supervises the geometric agreement between the tooth model and the actual tooth model by using the geometric spatial relation loss $L_{re}$, which is calculated as shown in Eq (16):

$$L_{re} = \frac{1}{n} \sum_{i=1}^{n} \left( \sum_{d=1}^{3} (p_{t,i,d} - g_{t,i,d})^2 \right) \cdot \omega_{tw,i} + \frac{1}{n} \sum_{i=1}^{n} \left( 1 - \left| \sum_{i=1}^{4} p_{r,i,d} \cdot g_{r,i,d} \right| \right) \tag{16}$$

where $p_{t,i,d}$ represents the target translation of the i-th tooth, $g_{t,i,d}$ represents the true translation of the i-th tooth, $\omega_{tw,i}$ is the weight, $p_{r,i,d}$ is the target quaternion, and $g_{r,i,d}$ is the true quaternion.

The proposed method is an end-to-end tooth position prediction method, and thus is jointly supervised by introducing a joint learning loss mechanism for tooth collision, point cloud reconstruction, and tooth transformation as in Eq (17):

$$L = \lambda_1 L_d + \lambda_2 L_{core} + \lambda_3 L_C + \lambda_4 L_{build} + \lambda_5 L_{dof} + \lambda_6 L_{re} \tag{17}$$

where $\lambda_1$ to $\lambda_5$ are the weights of the corresponding loss functions.

## Analysis of comparison and ablation experiments

### Dataset and implementation details

The original dataset of the proposed method is based on pathological characteristics and individual differences, and a total of 834 sets of case data are selected to form the dental position prediction dataset according to the ratio of 60% male and 40% female, and are divided into the training set and the test set according to the ratio of 8:2, in which the training set consists of 667 sets and the test set consists of 167 sets. In the physiological dataset construction stage, the GAOS strategy and the PARS strategy were used to construct the physiological dataset required by the proposed method, respectively. The experiments covered in this chapter are completed in Python environment, running on Ubuntu 20.04 system, GPU with NVIDIA RTX 4090 graphics card, 48G RAM, CPU with Intel Core i9 10900X, and deep learning framework with Pytorch 1.12.1+cu113.

The experiments' training parameters are set as follows: the training period is 4000 epochs, load data according to the specified path, read data according to the batch size of 8, and save

the model once every 1000 epochs of training. The optimizer was chosen to be Adam and dynamically adjusted using the cosine annealing learning rate scheduler. The loss function contains grid point distance loss, spatial core loss, chamfer distance loss, geometric reconstruction loss, geometric metric loss and geometric spatial relation loss. In addition, the model performance is optimized using CUDA and mixed precision training.

## Comparison and ablation of physiological dynamic optimization strategy

In order to evaluate the effectiveness of creating more diverse training samples that are compatible with clinical medicine through augmentation, as shown in Table 1, it can be found that the augmentation strategy has a significant effect in CSA, $ME_r$ and $ME_t$ by comparing the experiments of unaugmented GAOS and PARS. This is due to the different feature transformations and deep feature extraction techniques introduced by GAOS and PARS strategies in the optimization process. First, the GAOS and PARS strategies are able to capture the details and transformed features in the data more efficiently compared to the traditional Origin strategy. Specifically, the GAOS strategy makes the data not only retain the original feature information, but also mine the hidden and more representative features from different dimensions by introducing a dynamic intermediate transformation step of the model. In this way, the GAOS optimization strategy is able to diversify the data, and such optimization plays an important role in increasing the accuracy of the model. Unlike the GAOS strategy, the PARS strategy enables the model to better adapt to the physiological features of the data through specific physiological transformation techniques. When dealing with data with complex physiological characteristics, the PARS strategy enables the model to better understand the physiological patterns of change in the data by introducing physiological constraints and mathematical transformations of the oral cavity. This not only enhances the model's ability to capture physiologically relevant features, but also further optimizes the processing of the data through physiological transformations.

As shown in Table 2, the effectiveness of GAOS and PARS in this method is verified by comparing the IDR [14] and ITD [14] strategies. It can be clearly seen from Table 2 that the proposed method has a higher quality CSA because GAOS simulates the real transition process of the tooth from the initial to the final position, generates intermediate possible states, and increases the diversity of tooth position changes in the model training. The tooth positions generated by the tooth position prediction morphology generation strategy formed by introducing physiological constraints are more in line with the actual tooth movement patterns and oral structural limitations, which reduces the number of predictions that do not correspond to the real situation and increases the practicality and accuracy of the predictions.

In addition, the GAOS and PARS strategy utilizes physiological constraints to form a morphogenetic strategy for tooth position prediction that can more accurately reflect tooth movement patterns and oral structural limitations. With this strategy, the model-generated tooth

**Table 1. Ablation experiment of physiological dynamic optimization strategy.**

| Method | CSA↑ | | | $ME_r$ ↓ | $ME_t$ ↓ |
|---|---|---|---|---|---|
| | Mean | Minimum | Maximum | | |
| Origin | 0.7054 | 0.4851 | 0.8516 | 11.0406 | 1.4338 |
| GAOS | 0.8512 | 0.4875 | 0.9865 | 5.7939 | 0.9535 |
| PARS | 0.8713 | 0.4844 | 0.9879 | 5.2588 | 0.8652 |

Note: PARS and GAOS have better performance than unaugmented.

**Table 2. Comparative experiment of physiological dynamic optimization strategy.**

| Method | CSA↑ | | | ME$_r$ ↓ | ME$_t$ ↓ |
|--------|------|--------|---------|---------|---------|
| | Mean | Minimum | Maximum | | |
| **IDR** [14] | 0.8231 | 0.4822 | 0.9544 | 6.6650 | 1.0127 |
| **ITD** [14] | 0.8218 | 0.4884 | 0.9574 | 6.7674 | 1.0054 |
| **GAOS** | **0.8512** | **0.4875** | **0.9865** | **5.7939** | **0.9535** |
| **PARS** | **0.8713** | **0.4844** | **0.9879** | **5.2588** | **0.8652** |

**Note:** PCDPS and TDPS have better performance than IDR and ITD.

positions are more consistent with actual tooth movement patterns, avoiding unrealistic prediction results and increasing the usefulness and accuracy of the prediction results. Compared with the IDR and ITD strategies, the GAOS and PARS strategies not only consider single-dimensional rotations and displacements, but also incorporate the dynamic changes of the teeth, which can effectively capture the subtle motions of the teeth in space, and avoid errors caused by ignoring these small changes when predicting tooth positions. These additional optimizations allow the GAOS and PARS strategies to demonstrate greater adaptability and higher accuracy when dealing with complex dental data. As a result, the proposed methods show significant advantages in generating tooth position prediction results that are more in line with the structural constraints of the actual oral cavity, especially in terms of effective improvement in practicality and accuracy.

## Comparison and ablation of F-VAMP networks

In order to validate the effectiveness of the network, the present method was analyzed in comparison with the tooth position prediction module in TANet [10] and the tooth position prediction module in jaw-tooth-landmark [9]. Thus, as shown in Table 3, the unit of $ME_t$ is in mm and the unit of $ME_r$ is in degrees. First, the backbone module was constructed, which extracts local features as well as global features from the tooth point cloud using encoders for each method. Secondly, the tooth position prediction module in TANet [10] and jaw-tooth-landmark [9] are reconstructed and the features extracted through the backbone module are input to the reconstructed tooth position prediction module by splicing to predict the transformation parameters for each tooth for alignment.

From Table 3, it can be seen that the F-VAMP tooth position prediction network of the proposed method obtained the highest CSA, and was also lower than the metrics of the other two tooth position prediction networks in $ME_r$ and $ME_t$. This is due to the fact that TANet mainly solves the problem of tooth position prediction for teeth with small differences in morphology between pre- and post-correction, and still suffers from inaccurate prediction of certain teeth with large variability when dealing with teeth with complex morphological

**Table 3. Comparative experiments of the tooth position prediction module.**

| Method | CSA↑ | | | ME$_r$ ↓ | ME$_t$ ↓ |
|--------|------|--------|---------|---------|---------|
| | Mean | Minimum | Maximum | | |
| *Base* | 0.6723 | 0.4827 | 0.8536 | 10.8439 | 1.7278 |
| *TANet* [10] | 0.8425 | 0.4722 | 0.9516 | 7.4042 | 1.3914 |
| *Jaw–Tooth–Landmark* [9] | 0.8613 | 0.5022 | 0.9624 | 5.8873 | 1.1446 |
| **Our** | **0.8767** | **0.5043** | **0.9928** | **5.2452** | **0.8338** |

**Note:** The units of $ME_r$ are degree; The units of $ME_t$ are millimeter; Our is the best way.

changes. Therefore, although TANet successfully extracts the features of teeth by deep learning network, the prediction accuracy of the model may be lower for highly deformed or special morphology teeth. In contrast, the proposed method refines the features of teeth with large morphological differences through the hierarchical feature module, which enables the model to fully consider the teeth with complex morphological changes in the learning process, and effectively improves the accuracy of tooth position prediction. Although Jaw-Tooth-Landmark guides the network for tooth position prediction by defining landmark constraints, its input tooth model lacks the comprehensive consideration of global and morphological features, and is more dependent on the dataset.

In addition, the tooth landmarks only focus on the contact points between teeth and fail to adequately capture more critical information that can help accurate prediction. Therefore, the proposed method effectively improves the learning ability of the model by extracting the basic features of the model surface as well as the local features of the tooth frames and axes, and combining the tooth apices, the connection points between the teeth and the contact points at the junction of the teeth and gingiva as the key feature markers, and this improvement effectively makes up for the deficiencies of the Jaw-Tooth-Landmark method and effectively improves the accuracy of the tooth position prediction.

The networks with different constraints are trained in order to verify the correctness and necessity of the jointly supervised network constraints in the proposed method approach. It can be seen from Table 4 that if the jointly supervised network constraints are removed, then the network degrades to having a single constraint on the geometrical relationship of the teeth, and the quality of the network decreases significantly. When the center-of-mass matching constraint, the chamfer distance constraint, and the collision prevention constraints are added as jointly supervised constraints, the proposed method verifies the validity of the jointly supervised constraints as shown in Table 4, and it is found that with the use of the center-of-mass matching constraint, by minimizing the difference between the centers of mass, it can be ensured that the source tooth center of mass is spatially and accurately aligned with the target tooth center of mass.

As shown in Fig 7, the test results of eight patients' teeth were selected from the results of the test patients, and by focusing on the visual results through the information in the box, it can be seen that the proposed method can better handle the collisions as well as misalignments of the teeth at the incisors and and molars compared to the compared methods. The reason for this is that the topological relationship between the teeth as well as the collision information is important information to be considered in the alignment process.

In Fig 8, the specific details of the visualization effect of tooth alignment prediction are highlighted. As shown in the first row of Fig 8, the tooth alignment can effectively avoid the occurrence of excessive gaps between teeth and misalignment of teeth, mainly due to the ability of the network-enhanced learning transformation matrix similarity of the proposed

**Table 4. Ablation experiments of the tooth position prediction module.**

| Method | CSA↑ | | | $ME_r$ ↓ | $ME_t$ ↓ |
|--------|------|---------|---------|---------|---------|
| | Mean | Minimum | Maximum | | |
| *Base* | 0.6723 | 0.4827 | 0.8536 | 10.8439 | 1.7278 |
| *F−VAMP* | 0.8678 | 0.4764 | 0.9720 | 5.4308 | 0.9039 |
| *F − VAMP + $L_c$* | 0.8702 | 0.4874 | 0.9780 | 5.2906 | 0.8877 |
| *F − VAMP + $L_d$* | 0.8730 | 0.4879 | 0.9837 | 5.6487 | 0.8637 |
| **Our** | **0.8767** | **0.5043** | **0.9928** | **5.2452** | **0.8338** |

Our is the best way.

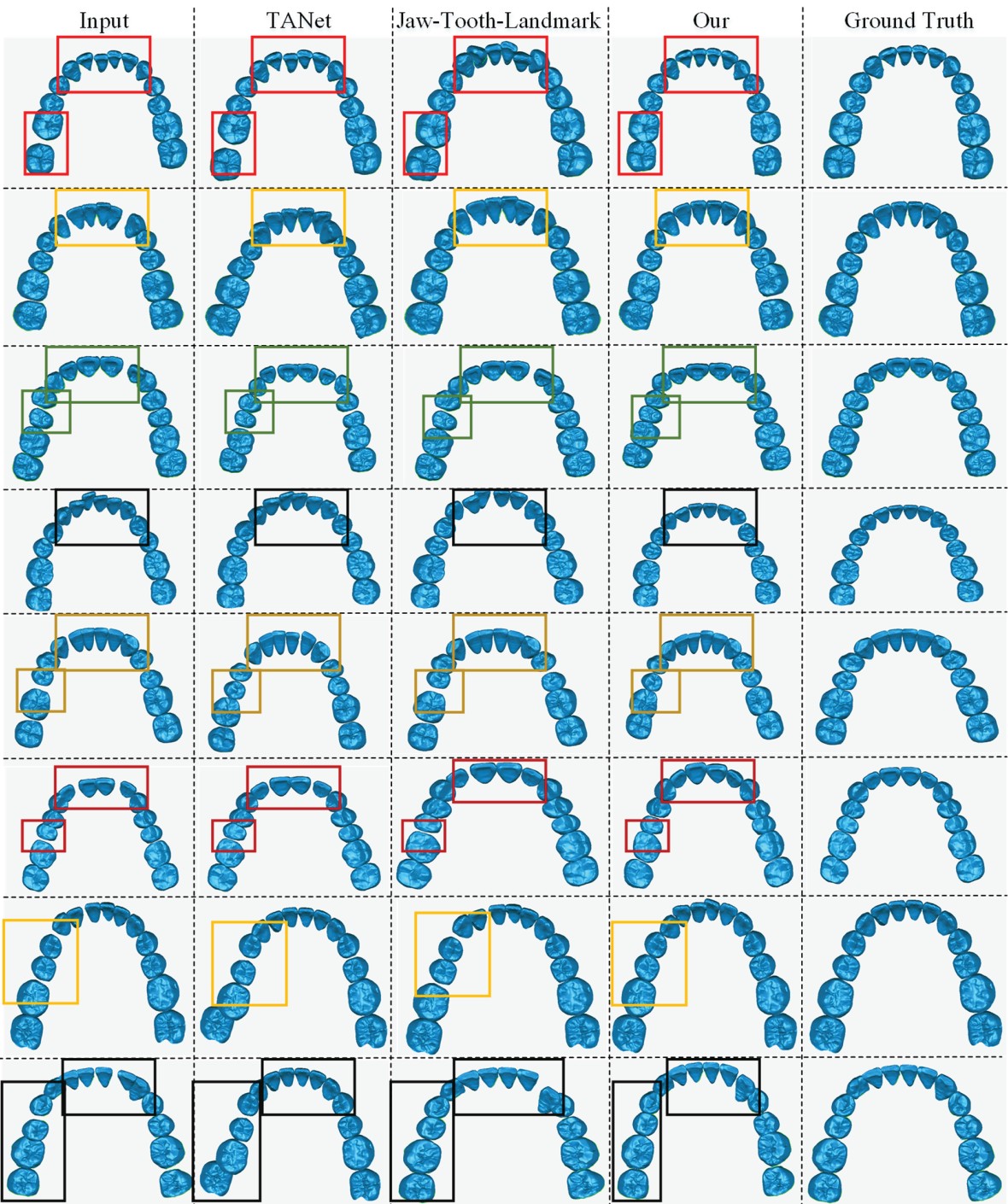

**Fig 7. Comparison of tooth position prediction results.** The regions highlighted by the rectangular boxes indicate the orthodontic areas.

method, and the intra- and inter-class similarity of teeth is obtained through the extraction of multi-layer features as well as the encoding of instance-aware information, which provides an effective help for the accuracy of tooth position placement and tooth connection. It can be concluded through the second line that the anti-collision loss and chamfer vector loss of

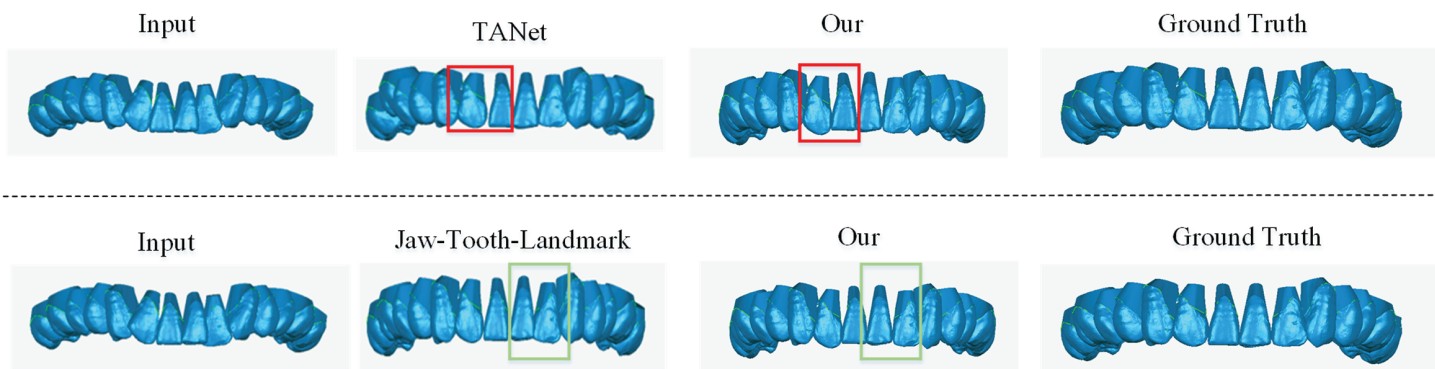

**Fig 8. Comparison of tooth collision and tooth misalignment.** The regions highlighted by the rectangular boxes indicate the areas where tooth misalignment and tooth collisions have been addressed.

the proposed method can help the teeth to extract the distance of the nearest neighbor points between the teeth and prevent the tooth collision situation as much as possible.

## Selection of key points

In order to evaluate the landmark selection in the proposed method, the validity of the proposed method network was verified by the arrangement of the landmarks of the four tooth types, as shown in Table 5, where a check mark indicates that the landmark is added to the network for tooth position prediction, and the result of the absence of the landmark selection indicates that the features of the landmarks of each tooth are abandoned and the landmarks of each tooth are deleted, where the first row indicates that the extracted module of the landmarks is completely removed The first line indicates the extraction module that completely removes the landmarks, followed by the test results of the tooth position prediction network in the case of different arrangements of landmarks for each tooth in turn.

**Table 5. Ablation experiments for landmark selection.**

| Landmark types | | | | CSA↑ | | | ME_r ↓ | ME_t ↓ |
|---|---|---|---|---|---|---|---|---|
| Incisors | Canines | Premolars | Molars | Mean | Minimum | Maximum | | |
| *No landmarks* | | | | 0.6723 | 0.4827 | 0.8536 | 10.8439 | 1.7278 |
| ✓ | | | | 0.8628 | 0.4850 | 0.9816 | 5.6619 | 0.8724 |
| | ✓ | | | 0.8642 | 0.4810 | 0.9817 | 5.5032 | 0.8779 |
| | | ✓ | | 0.8625 | 0.4880 | 0.9725 | 5.4592 | 0.8785 |
| | | | ✓ | 0.8147 | 0.4270 | 0.9662 | 8.6911 | 1.5204 |
| ✓ | ✓ | | | 0.8631 | 0.4889 | 0.9737 | 5.7830 | 0.8772 |
| ✓ | | ✓ | | 0.8620 | 0.4886 | 0.9816 | 5.8142 | 0.8932 |
| ✓ | | | ✓ | 0.8634 | 0.4852 | 0.9822 | 5.6975 | 0.8904 |
| | ✓ | ✓ | | 0.7832 | 0.4826 | 0.9261 | 6.6655 | 1.9207 |
| | ✓ | | ✓ | 0.8611 | 0.4864 | 0.9724 | 5.4085 | 0.8988 |
| | | ✓ | ✓ | 0.8631 | 0.4873 | 0.9810 | 5.4432 | 0.8639 |
| ✓ | ✓ | ✓ | | 0.8637 | 0.4840 | 0.9815 | 5.7025 | 0.8760 |
| ✓ | ✓ | | ✓ | 0.7490 | 0.3617 | 0.8946 | 8.2870 | 1.8797 |
| ✓ | | ✓ | ✓ | 0.8605 | 0.4872 | 0.9815 | 5.5589 | 0.8931 |
| | ✓ | ✓ | ✓ | 0.8210 | 0.4816 | 0.9624 | 7.1202 | 1.2816 |
| ✓ | ✓ | ✓ | ✓ | **0.8769** | **0.5075** | **0.9915** | **5.2324** | **0.8178** |

**Note:** All landmarks is the method that work best.

From the experimental results, it can be concluded that the proposed method achieves the highest accuracy when using the key points of each tooth since they emphasize the structural features of the tooth that are key to understanding the overall tooth arrangement and functional relationships, and the key points provide a clear reference to the exact spatial position of the tooth, which helps the network to better capture the exact position and orientation of each tooth with respect to other teeth.

## Conclusion

In this paper, we propose a tooth position prediction method based on adaptive geometric optimization, which solves the problem of lack of capturing spatial relationships and local correlations of teeth in the field of tooth position prediction using only Multilayer Perceptron machines. The experimental results show that the experimental results of this tooth position prediction method are better compared to the current tooth position prediction methods, and the experiments of the physiological dynamic optimization strategy are also better compared to the current strategy, and in terms of the landmarks, this method verifies that the prediction by the landmarks of each tooth is better than the prediction by the landmarks of a single tooth species, the landmarks of two tooth species, and the landmarks of three tooth species better. However, the study in this paper addresses single-maxillary tooth position prediction, and the interactions between the two jaws, coordination of maxillary and mandibular teeth, and joint modeling of two-maxillary movements have not yet been explored in depth, and the occlusal relationship and individual differences of the two jaws will pose greater challenges. Therefore, future studies need to further optimize the method so that it can effectively deal with the complex relationship between the two jaws, missing teeth, and individual differences, and optimize the bimaxillary dentition prediction.

## Author contributions

**Conceptualization:** Tian Ma, yijie zeng, Chao Li, Yuancheng Li.

**Data curation:** Tian Ma, yijie zeng, Wenda Pei.

**Formal analysis:** Tian Ma, yijie zeng, Wenda Pei.

**Funding acquisition:** Tian Ma, yijie zeng.

**Investigation:** Tian Ma, yijie zeng, Wenda Pei.

**Methodology:** Tian Ma, yijie zeng.

**Project administration:** Tian Ma, yijie zeng.

**Resources:** Tian Ma, yijie zeng.

**Software:** Tian Ma, yijie zeng.

**Supervision:** Tian Ma, yijie zeng.

**Validation:** Tian Ma, yijie zeng.

**Visualization:** Tian Ma, yijie zeng.

**Writing – original draft:** yijie zeng.

**Writing – review & editing:** Tian Ma, yijie zeng.

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
