## [Decision Letter · Decision Letter 0]

2 Apr 2025

PONE-D-24-46027Tooth Position Prediction Method based on Adaptive Geometry OptimizationPLOS ONE

Dear Dr. zeng,

Thank you for submitting your manuscript to PLOS ONE. After careful consideration, we feel that it has merit but does not fully meet PLOS ONE’s publication criteria as it currently stands. Therefore, we invite you to submit a revised version of the manuscript that addresses the points raised during the review process.

The manuscript could be improved by providing more detailed information on the dataset, computational resources, and statistical significance of the results. A more comprehensive review of related work and a critical discussion of limitations would enhance the manuscript's depth.

We look forward to receiving your revised manuscript.

Kind regards,

Alexander Maniangat Luke, PhD

Academic Editor

PLOS ONE

Journal Requirements:

Additional Editor Comments:

The introduction could benefit from a more detailed review of existing literature to better contextualize the novelty of the proposed method.

The methods could be enhanced by providing more information on the dataset used for training and testing, including the size and diversity of the data. Details on the computational resources and time required for training the model would be beneficial.

The results section could be strengthened by including more visualizations of the predicted tooth positions compared to the ground truth.

Statistical significance of the improvements should be discussed to reinforce the validity of the results.

The discussion could benefit from a more critical analysis of the limitations of the study and potential areas for future research. The impact of the proposed method on clinical workflows and patient outcomes could be elaborated.

The conclusion could be more forward-looking by suggesting specific future research directions or potential applications of the method in other areas of dental research.

Overall, the manuscript could be improved by providing more detailed information on the dataset, computational resources, and statistical significance of the results.A more comprehensive review of related work and a critical discussion of limitations would enhance the manuscript's depth.

Kindly address all comments of the reviewers.

Reviewers' comments:

Reviewer's Responses to Questions

**Comments to the Author**

1. Is the manuscript technically sound, and do the data support the conclusions?

Reviewer #1: Partly

2. Has the statistical analysis been performed appropriately and rigorously? 

Reviewer #1: No

3. Have the authors made all data underlying the findings in their manuscript fully available?

Reviewer #1: No

4. Is the manuscript presented in an intelligible fashion and written in standard English?

Reviewer #1: Yes

5. Review Comments to the Author

Reviewer #1: The manuscript presents a novel method for predicting tooth positions using adaptive geometry optimization. The approach introduces the Geometric Adaptive Optimization Strategy (GAOS) and the Physiological Adaptive Reconstruction Strategy (PARS) to improve accuracy in orthodontic modeling. Additionally, a hierarchical feature-based prediction network (F-VAMP) is proposed to mitigate the limitations of MLPs in high-dimensional data handling. The paper includes experimental results demonstrating improvements over existing methods.

Comments:

1. The introduction effectively presents the importance of the research problem, but it could benefit from a more concise overview of the existing methods and their limitations. Consider streamlining the discussion of prior works to avoid redundancy and emphasize the gap this paper addresses more explicitly.

2. The abstract is informative but overly detailed. Simplify technical jargon where possible, focusing on the core contributions and results. For instance, the introduction of specific methods such as GAOS and PARS could be summarized without extensive technical detail to improve readability.

3. The transition between sections is occasionally abrupt, particularly between the "Related Work" and "Methodology" sections. Adding bridging sentences to explain how the literature review informs the methodology would improve the coherence and flow.

4. Consider providing more detailed justifications for the choice of the physiological adaptive reconstruction strategy. While the use of physiological characteristics is mentioned, the connection to clinical relevance and practical orthodontic application could be more clearly articulated.

5. The manuscript lacks a clear explanation of the limitations and potential drawbacks of the proposed methods. It is crucial to acknowledge any challenges or potential issues to provide a balanced perspective and inform future research directions.

6. The figures, particularly those showing algorithmic flow (e.g., Figs 1 and 2), could be improved by adding clearer captions that explain their relevance in the context of the methodology. Captions should be more descriptive to guide readers unfamiliar with the concepts presented.

7. The mathematical formulations provided in the methodology are detailed, but there are occasional inconsistencies in notation and terminology. Ensure uniformity in the description of symbols across all equations, and clarify the definitions where symbols are first introduced.

8. Some sections are dense with technical language, which may hinder comprehension for a broader audience. For example, the physiological adaptive reconstruction strategy's description could benefit from simplified language or a supplementary explanation targeted at non-specialist readers.

9. The experimental results are promising, but the discussion section would benefit from a more thorough comparative analysis with state-of-the-art methods. Specifically, providing statistical significance testing of the improvements reported would add rigor and strengthen the validity of the findings.

10. The conclusion is well-structured but could be expanded to include a more explicit discussion of potential applications in clinical settings and future research opportunities that could further extend the work presented here.

The manuscript presents a significant contribution to the field of orthodontic modeling. However, improvements in clarity, structure, and a more balanced discussion of limitations are necessary for the manuscript to be considered for publication. Adding more explicit justification and context for the proposed methods would also enhance the manuscript’s impact.

6. PLOS authors have the option to publish the peer review history of their article (what does this mean?). If published, this will include your full peer review and any attached files.

Reviewer #1: **Yes: **Mehrad Aria

---

## [Author Response · Author response to Decision Letter 1]

24 Apr 2025

Answer all comments of the reviewers

1.The introduction effectively presents the importance of the research problem, but it could benefit from a more concise overview of the existing methods and their limitations. Consider streamlining the discussion of prior works to avoid redundancy and emphasize the gap this paper addresses more explicitly.

Answer: Dear Expert, I have simplified the introduction section, with the red-highlighted portions reflecting the revised content. The main change involves streamlining the discussion of previous work, and reducing it by approximately half.

2.The abstract is informative but overly detailed. Simplify technical jargon where possible, focusing on the core contributions and results. For instance, the introduction of specific methods such as GAOS and PARS could be summarized without extensive technical detail to improve readability.

Answer: Dear Expert, I have simplified the technical terms related to GAOS and PARS in the red-highlighted sections of the abstract to make it more focused on the core contributions and results. The main revisions are located in lines 4 to 7 of the abstract.

3. The transition between sections is occasionally abrupt, particularly between the "Related Work" and "Methodology" sections. Adding bridging sentences to explain how the literature review informs the methodology would improve the coherence and flow.

Answer: Dear Expert, I have added bridging sentences in lines 1 to 14 of the third paragraph of the related work section to improve the transition. The changes have been highlighted in red.

4. Consider providing more detailed justifications for the choice of the physiological adaptive reconstruction strategy. While the use of physiological characteristics is mentioned, the connection to clinical relevance and practical orthodontic application could be more clearly articulated.

Answer: Dear Expert, I have added the clinical relevance and connection to practical orthodontic applications for the Physiological Adaptive Reconstruction Strategy. The main revisions can be found in lines 5 to 11 of the first paragraph, highlighted in red.

5. The manuscript lacks a clear explanation of the limitations and potential drawbacks of the proposed methods. It is crucial to acknowledge any challenges or potential issues to provide a balanced perspective and inform future research directions.

Answer: Dear Expert, I have added an explanation of the limitations, potential drawbacks and the scope for further optimization of the proposed method in the conclusion section. The revisions can be found in lines 11 to 17 of the conclusion section, it has highlighted in red.

6. The figures, particularly those showing algorithmic flow (e.g., Figs 1 and 2), could be improved by adding clearer captions that explain their relevance in the context of the methodology. Captions should be more descriptive to guide readers unfamiliar with the concepts presented.

Answer: Dear Expert, I have revised the titles of the figures for better clarity. The title of Fig 1 has been changed from "GAOS Diagram" to "Effect of GAOS Strategy to Generate Dataset" and the title of Fig 2 has been changed from "PARS Diagram" to "Effect of PARS Strategy to Generate Dataset" in order to better reflect their functions and concepts for easier understanding by the readers.

7. The mathematical formulations provided in the methodology are detailed, but there are occasional inconsistencies in notation and terminology. Ensure uniformity in the description of symbols across all equations, and clarify the definitions where symbols are first introduced.

Answer: Dear Expert, after revisiting the details of the mathematical formulas, I have corrected the consistency in the symbol descriptions and provided explanations for symbols introduced for the first time. The main revisions are in Eq(4)–(6), (8)–(9) and (12)–(16), with the changes have highlighted in red.

8. Some sections are dense with technical language, which may hinder comprehension for a broader audience. For example, the physiological adaptive reconstruction strategy's description could benefit from simplified language or a supplementary explanation targeted at non-specialist readers.

Answer: Dear Expert, I have simplified the technical language of the Physiological Adaptive Reconstruction Strategy. The main revisions are in lines 1 to 4 of the first paragraph, and it has highlighted in red to facilitate easier reading for non-specialist readers.

9. The experimental results are promising, but the discussion section would benefit from a more thorough comparative analysis with state-of-the-art methods. Specifically, providing statistical significance testing of the improvements reported would add rigor and strengthen the validity of the findings.

Answer: Dear Expert, the method is compared with the physiological dynamic optimization strategy in this study is the one published in 2020 in the International Conference on Medical Image Computing and Computer-Assisted Intervention. This paper is currently the only one in the field of tooth position prediction that proposes an optimization strategy specifically for datasets, which is why I selected it as the comparison method.

The methods compared with the F-VAMP network are published in 2022 on the IEEE Transactions on Visualization and Computer Graphics and in 2020 on the European Conference on Computer Vision. These papers are of high academic quality in the field of tooth position prediction and are published recently. Additionally, most of the recent literature on orthodontics focuses on tooth arrangement rather than tooth position prediction. Therefore, this study selects these papers as comparison methods for the experimental analysis of tooth position prediction.

Furthermore, the statistical significance metrics were obtained by training and testing datasets derived from 834 original data sets, which were split in an 8:2 ratio, using the GAOS and PARS strategies. I have conducted a more thorough comparative analysis of the methods being compared, including the innovative technical details of the GAOS and PARS strategies, as well as a comparative evaluation of these strategies.

On the other hand, the comparison analysis of F-VAMP provides a more in-depth evaluation of both the shortcomings of the comparison methods and the advantages of the proposed method. The revisions are highlighted in red in the sections Comparison and Ablation of Physiological Dynamic Optimization Strategy and Comparison and Ablation of F-VAMP Networks.

10.The conclusion is well-structured but could be expanded to include a more explicit discussion of potential applications in clinical settings and future research opportunities that could further extend the work presented here.

Answer: Dear Expert, I have made revisions in the Conclusion section, specifically in lines 1 to 10 and 15 to 17, where the red-highlighted text now provides a clearer discussion of the current research context and future research opportunities. Additionally, I have further elaborated on the contributions of the proposed method.

Answer all Additional Editor Comments:

1.The introduction could benefit from a more detailed review of existing literature to better contextualize the novelty of the proposed method.

Answer: Dear Expert, I have provided a more detailed review of the existing literature, with specific revisions highlighted in red. These changes can be found in the second paragraph of the related work section, in lines 2 to 11 and 17 to 23.

2. The methods could be enhanced by providing more information on the dataset used for training and testing, including the size and diversity of the data. Details on the computational resources and time required for training the model would be beneficial.

Answer: Dear Expert, I have provided additional information on the size and diversity of the training and testing datasets to strengthen these methods, along with a detailed description of the experimental setup. The specific details can be found in the Dataset and Implementation Details section and are highlighted in red.

3. The results section could be strengthened by including more visualizations of the predicted tooth positions compared to the ground truth.

Answer: The statistical metrics include the quantified translations and rotations of the predicted positions. Additionally, two sets of visualizations illustrating the experimental effects have been added to Fig. 7. Due to space limitations in Fig. 7, the quantified data regarding the predicted positions and ground truth are provided in Table 3.

4. Statistical significance of the improvements should be discussed to reinforce the validity of the results.

Answer: Dear Expert, The statistical significance metrics were obtained by training and testing datasets derived from 834 original data sets, split in an 8:2 ratio, using the GAOS and PARS strategies. Additionally, I have conducted a more thorough comparative analysis of the methods being compared, with a particular focus on the innovative technical details of the GAOS and PARS strategies, as well as a detailed comparison between the two.

Furthermore, the comparison analysis of F-VAMP provides a more in-depth evaluation of both the shortcomings of the comparison methods and the advantages of the proposed method. The revisions are highlighted in red.

5. The discussion could benefit from a more critical analysis of the limitations of the study and potential areas for future research. The impact of the proposed method on clinical workflows and patient outcomes could be elaborated.

Answer: Dear Expert, I have added an explanation of the limitations, potential drawbacks, and the scope for further optimization in future research regarding the proposed method in the Conclusion section. The revisions can be found in lines 11 to 17 of the Conclusion, highlighted in red.

6. The conclusion could be more forward-looking by suggesting specific future research directions or potential applications of the method in other areas of dental research.

Answer: Dear Expert, I have made revisions in the Conclusion section, specifically in lines 1 to 10 and 15 to 17, where the red-highlighted text now offers a clearer discussion of the current research context and future research opportunities. Additionally, I have further elaborated on the contributions of the proposed method.

7.Overall, the manuscript could be improved by providing more detailed information on the dataset, computational resources, and statistical significance of the results.A more comprehensive review of related work and a critical discussion of limitations would enhance the manuscript's depth.

Answer: Dear Expert, I have provided the code and sample dataset for this study at the following URL: https://gitcode.com/zyj_jj/TPPMAGO. All datas are stored in URL: https://gitcode.com/zyj_jj/TPPMAGO_toothdata/blob/main/README.md

Improvements have been made to the statistical significance of the physiological dynamic optimization strategy and the F-VAMP network, with a more thorough analysis from multiple perspectives. More detailed information on the dataset and computation methods is provided in the Dataset and Implementation Details section.

Additionally, I have conducted a more comprehensive review of the related work and provided a critical discussion of the limitations. The specific revisions are highlighted in red in the second paragraph of the Related Work section, in lines 2 to 11 and 17 to 23.

All Journal Requirements:

1.Please ensure that your manuscript meets PLOS ONE's style requirements, including those for file naming. 

Answer: Dear Editor, this manuscript has been written strictly according to the latest LaTeX template provided on the PLOS ONE website, and the formatting has been adjusted in accordance with the submission guidelines specified on the PLOS ONE website.

2.Please note that PLOS ONE has specific guidelines on code sharing for submissions in which author-generated code underpins the findings in the manuscript. In these cases, we expect all author-generated code to be made available without restrictions upon publication of the work. Please review our guidelines at https://journals.plos.org/plosone/s/materials-and-software-sharing#loc-sharing-code and ensure that your code is shared in a way that follows best practice and facilitates reproducibility and reuse.

Answer: Dear Editor, the code for this manuscript can be accessed at the following URL: https://gitcode.com/zyj_jj/TPPMAGO, where the dataset folder contains the sample dataset. All the datasets are stored in the following link: https://gitcode.com/zyj_jj/TPPMAGO_toothdata/blob/main/README.md

Answer: Dear expert, I have uploaded the latest Financial Disclosure Statement report as "Other" in the online submission of the Attach Files system. To ensure that the "Funding Information" matches the "Financial Disclosure".

Answer: Dear Editor, I have made the code publicly available, and the corresponding URL is: https://gitcode.com/zyj_jj/TPPMAGO, where the dataset folder contains the sample dataset. All datas are stored in URL: https://gitcode.com/zyj_jj/TPPMAGO_toothdata/blob/main/README.md

Note: Dear Editor, due to a change in the name of my institution in April 2025, I have updated the name of the college from the former "College of Computer Science and Technology" to the new "College of Artificial Intelligence & Computer Science."

---

## [Decision Letter · Decision Letter 1]

23 May 2025

PONE-D-24-46027R1Tooth Position Prediction Method based on Adaptive Geometry OptimizationPLOS ONE

Dear Dr. zeng,

Thank you for submitting your manuscript to PLOS ONE. After careful consideration, we feel that it has merit but does not fully meet PLOS ONE’s publication criteria as it currently stands. Therefore, we invite you to submit a revised version of the manuscript that addresses the points raised during the review process.

We look forward to receiving your revised manuscript.

Kind regards,

Hakan Turkkahraman

Academic Editor

PLOS ONE

Journal Requirements:

Reviewers' comments:

Reviewer's Responses to Questions

**Comments to the Author**

1. If the authors have adequately addressed your comments raised in a previous round of review and you feel that this manuscript is now acceptable for publication, you may indicate that here to bypass the “Comments to the Author” section, enter your conflict of interest statement in the “Confidential to Editor” section, and submit your "Accept" recommendation.

Reviewer #1: All comments have been addressed

Reviewer #2: (No Response)

2. Is the manuscript technically sound, and do the data support the conclusions?

Reviewer #1: Yes

Reviewer #2: Partly

3. Has the statistical analysis been performed appropriately and rigorously? 

Reviewer #1: Yes

Reviewer #2: No

4. Have the authors made all data underlying the findings in their manuscript fully available?

Reviewer #1: Yes

Reviewer #2: Yes

5. Is the manuscript presented in an intelligible fashion and written in standard English?

Reviewer #1: Yes

Reviewer #2: No

6. Review Comments to the Author

Reviewer #1: (No Response)

Reviewer #2: Main contents:

This work proposes a transformer architecture for tooth position prediction in digital orthodontics. The authors proposed the geometric adaptive optimization strategy (GAOS) and physiological adaptive reconstruction strategy (PARS) to construct physiological tooth dataset. These data augmentation strategies can improve the tooth position prediction performance of the proposed transformer architecture.

Main weakness:

1. The English writing of the manuscript needs to be significantly improved. There are many typos in the current manuscript. For example, in lines 93-94, two sentences are repeated. Grammar mistakes should be corrected to make the manuscript readable.

2. Many repeated details of the proposed two strategies and transformer architecture should be omitted to make expressions clear and easy to read.

3. The presentation of the proposed method and experimental results should be improved. For example, authors can enrich the captions and illustrations under each figure.

7. PLOS authors have the option to publish the peer review history of their article (what does this mean?). If published, this will include your full peer review and any attached files.

Reviewer #1: **Yes: **Mehrad Aria

Reviewer #2: No

---

## [Author Response · Author response to Decision Letter 2]

12 Jun 2025

Answer all comments of the reviewers

1.The English writing of the manuscript needs to be significantly improved. There are many typos in the current manuscript. For example, in lines 93-94, two sentences are repeated. Grammar mistakes should be corrected to make the manuscript readable.

Answer: Regarding the issue, we have removed the redundant sentence in lines 93-94. Furthermore, we have checked the entire manuscript, corrected all the typos errors and improved the English writing. These modifications are specifically reflected in red, such as lines 103-105, 119, 176-177, 192-200, 229, 239, 386, 419, 426-435, 446, and 450-452, and so on.

2.Many repeated details of the proposed two strategies and transformer architecture should be omitted to make expressions clear and easy to read.

Answer: I have omitted many of the repetitive details regarding the proposed two strategies and the Transformer. The modifications to the GAOS strategy are reflected in lines 124-131, the changes to the PARS strategy are reflected in lines 149-156, and the adjustments to the Transformer architecture are shown in lines 210-221. The specific changes have been highlighted in red text.

3.The presentation of the proposed method and experimental results should be improved. For example, authors can enrich the captions and illustrations under each figure.

Answer: The content of Figures 1 to 8 has been enriched in this paper. The specific modifications are located after the titles of Figures 1 to 8(between lines 131-132, 168-169, 226-227, 243-244, 267-268, 287-288, and 424-425), and the changes have been highlighted in red text.

All Journal Requirements:

Answer: I have checked the status of each reference cited in this paper through Google Scholar, and all of them are published, complete and correct references.

---

## [Editor Report · Decision Letter 2]

17 Jun 2025

基于自适应几何优化的齿位置预测方法

PONE-D-24-46027R2

Dear Dr. zeng,

We’re pleased to inform you that your manuscript has been judged scientifically suitable for publication and will be formally accepted for publication once it meets all outstanding technical requirements.

Kind regards,

Hakan Turkkahraman

Academic Editor

PLOS ONE

---

## [Editor Report · Acceptance letter]

PONE-D-24-46027R2

PLOS ONE

Dear Dr. zeng,

I'm pleased to inform you that your manuscript has been deemed suitable for publication in PLOS ONE. Congratulations! Your manuscript is now being handed over to our production team.

Kind regards,

on behalf of

Dr. Hakan Turkkahraman

Academic Editor

PLOS ONE